# Contrast-Enhanced Ultrasound Qualitative and Quantitative Characteristics of Parathyroid Gland Lesions

**DOI:** 10.3390/medicina58010002

**Published:** 2021-12-21

**Authors:** Sergejs Pavlovics, Maija Radzina, Rita Niciporuka, Madara Ratniece, Madara Mikelsone, Elina Tauvena, Mara Liepa, Peteris Prieditis, Arturs Ozolins, Janis Gardovskis, Zenons Narbuts

**Affiliations:** 1Radiology Research Laboratory, Riga Stradins University, LV-1007 Riga, Latvia; maija.radzina@rsu.lv (M.R.); ratniece.madara@gmail.com (M.R.); mara.liepa@rsu.lv (M.L.); peteris.prieditis@rsu.lv (P.P.); 2Diagnostic Radiology Institute, Pauls Stradins Clinical University Hospital, LV-1002 Riga, Latvia; 3Faculty of Medicine, University of Latvia, LV-1586 Riga, Latvia; 4Department of Surgery, Riga Stradins University, LV-1007 Riga, Latvia; rita.niciporuka@gmail.com (R.N.); arturs.ozolins@stradini.lv (A.O.); janis.gardovskis@stradini.lv (J.G.); zenons.narbuts@stradini.lv (Z.N.); 5Department of Surgery, Pauls Stradins Clinical University Hospital, LV-1002 Riga, Latvia; 6Faculty of Medicine, Riga Stradins University, LV-1007 Riga, Latvia; tauvenaelina@gmail.com; 7Department of Statistics, Riga Stradins University, LV-1007 Riga, Latvia; madara.mikelsone@rsu.lv

**Keywords:** hyperparathyroidism, parathyroid adenoma, parathyroid hyperplasia, contrast-enhanced ultrasound, scintigraphy

## Abstract

*Background and Objectives*: preoperative differentiation of enlarged parathyroid glands may be challenging in conventional B-mode ultrasound. The aim of our study was to analyse qualitative and quantitative characteristics of parathyroid gland lesions, using multiparametric ultrasound protocol—B-mode, Colour Doppler (CD), and contrast-enhanced ultrasound (CEUS)—and to evaluate correlation with morphology in patients with hyperparathyroidism (HPT). *Materials and Methods*: consecutive 75 patients with 88 parathyroid lesions and biochemically confirmed HPT prior to parathyroidectomy were enrolled in the prospective study. B-mode ultrasound, CD, and CEUS were performed with the subsequent qualitative and quantitative evaluation of acquired data. We used 1 mL or 2 mL of intravenous ultrasound contrast agent during the CEUS examination. Correlation with post-surgical morphology was evaluated. *Results:* seventy parathyroid adenomas were hypoechoic and well contoured with increased central echogenicity (44.3%), peripheral-central vascularization (47%), and polar feeding vessel (100%). Twelve hyperplasias presented with similar ultrasound appearance and were smaller in volume (*p* = 0.036). Hyperplasias had a tendency for homogenous, marked intense enhancement vs. peripherally enhanced adenomas with central wash-out in CEUS after quantitative analysis. No significant difference was observed in contrasting dynamics, regardless of contrast media volume use (1 mL vs. 2 mL). We achieved 90.9% sensitivity and 72.7% specificity, 93% positive predictive value (PPV), 87.3% negative predictive value (NPV), and 87.3% accuracy in the differentiation of parathyroid lesions prior to post-processing. In a quantitative lesion analysis, our sensitivity increased up to 98%, specificity 80%, PPV 98%, and NPV 80% with an accuracy of 96.4%. *Conclusions:* CEUS of parathyroid lesions shows potential in the differentiation of adenoma from hyperplasia, regardless of the amount of contrast media injected. The quantitative analysis improved the sensitivity and specificity of differentiation between parathyroid lesions. Hyperplasia was characterized by homogeneous enhancement, fast uptake, and homogeneous wash-out appearance; adenoma—by peripheral uptake, central wash-out, and reduced hemodynamics. The use of CEUS quantification methods are advised to improve the ultrasound diagnostic role in suspected parathyroid lesions.

## 1. Introduction

There are three types of hyperparathyroidism: primary, secondary, and tertiary. Primary hyperparathyroidism (HPT) is the third most common endocrine disorder, after diabetes and thyroid diseases, and the leading cause of hypercalcemia [1]. The prevalence of this disorder is estimated between 0.4–82 cases per 100.000 population, more commonly among women [2]. The prevalence of primary HPT, in combination with thyroid pathologies, varies from 17% to 84%, depending on the study population [1].

Solitary parathyroid adenoma (PA) is the most common cause of primary HPT (80%), followed by multiglandular disease (multiple adenomas or hyperplasias—20%) and carcinoma (<1%) [2]. Parathyroid hyperplasia (PH) is defined by the increase in parenchymal cell mass [3]. Secondary and tertiary HPT may represent as kidney failure, severe vitamin D deficiency, and other conditions, leading to calcium loss [4]. Symptomatic patients may present with osteopenia, osteoporosis, fatigue, kidney stones, and calcification of heart valves, among other symptoms [5].

Ultrasonography (US) is one of the main imaging modalities for the evaluation of parathyroid diseases. It may be challenging to differentiate enlarged parathyroid glands from thyroid nodules and lymph nodes in B-mode US because of visual similarities between these structures, especially in goitres and after previous neck surgery [6,7,8]. Preoperative parathyroid localization in coexisting thyroid pathology is known to have low sensitivity and specificity [1]. Doppler US may be helpful as PA, and hyperplasia’s are usually hypervascular, and they usually have a prominent extra thyroidal feeding artery entering the pole and then extending around the periphery of the enlarged gland [9,10]. The feeding polar vessel is particularly useful in differentiating a PA from a cervical lymph node, where the feeding vessel enters the hilum centrally. However, this finding is not common in all parathyroid lesions [9]. The sensitivity of conventional US (B-mode and Color Doppler US (CD)) is particularly low when PA is associated with goitres. To improve the accuracy of conventional US, contrast-enhanced US (CEUS), with intravenous administration of a contrast agent, is being advised [6,11].

The contrast agent used during CEUS examination enhances reflections of vascular signals received from lesions [12]. This enables detection of tissue microvascularisation during CEUS and facilitates differentiation between enlarged parathyroid glands, thyroid nodules, and lymph nodes. The use of CEUS is approved by the European Federation of Societies for Ultrasound in Medicine and Biology (EFSUMB) guidelines as a safe method for both adults and children [13,14].

Surgery remains the only definitive treatment for primary HPT [15]. Disease recognition, a proper diagnostic process, and preoperative preparation of the patients are crucial steps for choosing the best surgical approach. The most appropriate type of surgical intervention depends on the number and localization of the hyperactive parathyroid glands [16,17].

The aim of our study was to analyse qualitative and quantitative characteristics of parathyroid gland lesions using multiparametric US protocol (B-mode, CD, and CEUS) and to evaluate correlation with morphology in patients with HPT.

## 2. Materials and Methods

### 2.1. Study Design and Population

There were 75 patients, aged 19–82 years, with HPT scheduled for parathyroidectomy enrolled in our study, and a total of 88 morphological samples were obtained since May 2019 to October 2021 (Table 1, Figure 1).

### 2.2. Study Methodology

Patients underwent multiparametric US evaluation of thyroid, parathyroid, and neck soft tissues prior to parathyroid surgery within routine patient management and standard pre-treatment imaging protocols for parathyroid pathology assessment in our hospital.

The examinations were performed using the Aplio i800 system (Canon, Japan). All subjects were positioned supine with the neck slightly extended, and an adequate amount of gel was applied in the lower neck area. All examinations were performed by a radiologist with more than ten years of experience. Patients were asked to their hold breath during the examination to avoid motion artefacts.

The visual appearance of the thyroid gland and enlarged parathyroid glands were evaluated in transverse and longitudinal planes, using the conventional US. CEUS was performed after conventional US examination, the mechanical index was reduced to less than 0.10, and the parallel B-mode US and CEUS imaging was obtained. Twenty-nine lesions were evaluated using 1 mL of contrast agent, 59 lesions were evaluated using 2 mL of contrast agent SonoVue (Bracco, Milan) intravenously, followed by saline flush (10 mL). CEUS scanning was performed continuously for the first 60 s and was carried out intermittently for up to 2 min. The dynamic process of scanning was stored as cine loops and still images with information about timing. Stored clips were later reviewed, as the examination is dynamic, and subtle uptake may require additional analysis. Images and cine loop material were obtained and saved in the hospital’s picture archiving and communication system (PACS).

Additionally, 99mTc-MIBI scintigraphy (planar scintigraphy 13.4% and SPECT/CT 86.6% with dual phase, respectively) was performed in 60 (80%) patients out of 75.

### 2.3. Qualitative Parameters

Qualitative parameters were evaluated in 82 parathyroid lesions. Qualitative parameters of multiparametric US were: size in millimetres (mm), volume (cubic millimetres, mm^3^), and echogenicity of a lesion (hyperechoic/isoechoic/hypoechoic compared to thyroid) structural homogeneity and presence of cystic inclusions within a lesion. CD was used to assess vascularisation patterns in lesions (central/peripheral/combined) and to detect the presence of one polar feeding vessel.

Qualitative CEUS parameters were used to analyse enhancement parameters (presence of contrast uptake, time of uptake, time of maximal concentration, presence of wash-out, wash-out time) and contrast dynamics patterns of parathyroid lesions (predominantly central/peripheral vs. homogenous). Laboratory findings of serum parathormone (reference range 18.5–88 pg/mL) and serum calcium (reference range 2.10–2.60 mmol/L) were compared in patients with PA and PH.

Qualitative multiparametric US data and laboratory (blood calcium and blood parathormone) findings were compared with morphology (adenoma, hyperplasia) results.

### 2.4. Quantitative Parameters

Quantitative parameters were evaluated in 51/82 lesions (62.1%). Post-processing of acquired CEUS data using VueBox (Bracco, Italy) application was performed for in-depth quantitative evaluation. We created regions of interest (ROI) in the parathyroid lesions central part (green), periphery (orange), and in the thyroid (red) (Figure 2).

Quantitative analysis of selected ROI was performed, and Average Contrast Signal Intensity (ACSI), Wash-in rate (WiR), and Wash-out rate (WoR) were analysed. ACSI, within a region of interest (ROI), can be displayed as a function of time, in the form of a time-intensity curve (TIC), which describes the wash-in and wash-out of the contrast agent in the ROI, ACSI is expressed as echo-power, in a relative units (arbitrary units, a.u.) (Figure 3) [18,19]. Thyroid (red) analysis was later excluded, as some patients have had thyroidectomy or presented with atypical parathyroid lesion localisation, thus, the thyroid gland could not be included in the field of view.

Patients with single PA were divided into two groups—those with PA volume less than 1000 mm^3^ (Group 1) and those with PA volume equal to or above 1000 mm^3^ (Group 2)—with the comparable count in both groups to analyse lesion size impact on CEUS appearance.

### 2.5. Surgery

According to preoperative laboratory results, radiological findings, and surgical guidelines patients underwent focused parathyroidectomy or unilateral/bilateral parathyroid exploration with consecutive parathyroidectomy. Intraoperative parathormone measurement and next-day calcium and parathormone levels monitoring were performed to confirm successful parathyroidectomy. Parathyroid specimens were sent to the pathologist for a routine morphological description of parathyroid tissue. Parathyroidectomy was performed by an experienced team of endocrine surgeons, and the final diagnosis was based on morphology results.

### 2.6. Statistical Analysis

Statistical evaluation was performed using IBM SPSS 28.0 (IBM Corp., Armonk, NY, USA). Categorical variable comparisons were performed using the chi-square or Pearson exact test, and continuous variables were evaluated with the Student *t*-test or Mann–Whitney U-test, as appropriate. A *p*-value < 0.05 was used to define significant results. Analysis of variance (ANOVA) was used to compare means of all parameters within groups: adenoma or hyperplasia.

## 3. Results

### 3.1. Qualitative Parameters

We were able to evaluate 82 of 88 surgically removed parathyroid lesions (93.2%) in our study, using the conventional US. PH were smaller compared to PA (*p* = 0.039). The majority of PH were in a group of <1000 mm^3^. There was a similar amount of PA in both groups: <1000 mm^3^ and ≥1000 mm^3^. All lesions were hypoechoic. Some PH and PA presented, with hyperechoic central parts, with no significant difference (*p* = 0.70). A partial cystic component was more frequent in PHs, compared to PAs (*p* = 0.01). The most common combined vascularisation pattern (central and peripheral) was observed in both PH and PA (Table 2). Single PA qualitative analysis is represented in Table 3. Larger PA ≥ 1000 mm^3^ had higher parathormone (*p* = 0.005) and slightly higher calcium blood levels. The onset of contrast uptake, time of maximum contrast concentration, and onset of wash-out tended to be more rapid in a smaller volume lesion group with no statistical significance.

### 3.2. Quantitative Parameters

In quantitative analysis, ACSI was similar in lesions where 1 mL of contrast agent was injected during the US, compared to 2 mL in central and peripheral parts (*p* = 0.386 and *p* = 0.436, respectively). There was no difference observed in wash-in and wash-out rates of parathyroid lesion central (r = −0.108, *p* = 0.402 and r = −0.125 *p* = 0.436, respectively) and peripheral (r = −0.109, *p* = 0.631 and r = −0.147, *p* = 0.356, respectively) parts in patients who received 1 mL vs. 2 mL of contrast agent (Table 4). ACSI was higher in larger lesions, both in the centre (*p* = 0.406) and periphery (*p* = 0.503).

On individual time intensity curves, PA had lower ACSI in the central part compared to the periphery. PH had higher and similar ACSI values in central and peripheral parts, compared to PA (Figure 4). Wash-in and wash-out rate was similar in both the centre (r = 0.139, *p* = 0.413 and r = 0.150, *p* = 0.383, respectively) and periphery (r = 0.068, *p* = 0.624 and r = 0.133, *p* = 0.448, respectively) in lesions below and above 1000 mm^3^ (Figure 5).

Higher ACSI was observed in PH’s, compared to PA’s, both in lesion central (r = −0.425, *p* = 0.002) and peripheral (r = −0.495, *p* = 0.001) parts. PH presented with higher wash-in and wash-out rates in central and peripheral parts, compared to PA (*p* = 0.001). Slightly higher ACSI, and the wash-in rate, were observed in peripheral parts of PH and PA (Table 5, Figure 6 and Figure 7).

All patients in our study that could be detected on B-mode ultrasound presented with hypercalcemia, and thus, the overall sensitivity of CEUS in a whole group of patients with hypercalcemia, due to hyperparathyroidism, was 90.9% prior to quantitative analysis. There were three patients presenting with normocalcemia, and B-mode US did not reveal lesions, although parathormone levels were elevated. There were no specific contrast-enhancement patterns observed that could be related to parathormone values.

Parathyroid volumes and largest diameter showed no significant correlation with CEUS findings (r*_p_* = 0.006; *p* = 0.95), showing no difference among hyperplasia and adenoma. Association with volume for hyperplasia vs. adenoma <2000 mm^3^ was 86.7% vs. 74.3% and for ≥2000 mm^3^—13.3% vs. 25.7% (*p* = 0.3), respectively. The sensitivity of CEUS, depending on lesion volume at histology, was 0.64% (CI 0.51–0.76; *p* = 0.03). Sensitivity had decreasing tendency in smaller lesions (median 374 mm^3^ (183–1547 mm^3^)) vs. larger lesions (median 916 mm^3^ (282–1893 mm^3^)).

Correct localisation was determined in 80/88 (90.9%) parathyroid lesions. Three lesions could not be seen in the US, and an incorrect quadrant (upper vs. lower) was reported in five cases. Incorrect morphology (adenoma vs. hyperplasia) was determined in 16/88 cases (18.2%). We achieved 90.9% sensitivity and 72.7% specificity, 93% positive predictive value (PPV), 87.3% negative predictive value (NPV), and 87.3% accuracy in diagnosing parathyroid lesions prior to post-processing. After quantitative lesion analysis, our sensitivity was 98%, specificity 80%, PPV 98%, NPV 80%, and accuracy was 96.4%.

The sensitivity of 99mTc-MIBI (planar scintigraphy and SEPCT/CT) alone was 68.4% (CI 0.513–0.825; *p* = 0.7), but when combined with CEUS ultrasound, the sensitivity reached up to 94.3% (CI 0.518–0.841; *p* = 0.03).

## 4. Discussion

It may be challenging to differentiate adenoma (PA) from hyperplasia (PH) and other neck structures, such as lymph nodes and thyroid nodules, because of visual similarities of these structures on the conventional US [21,22]. Both parathyroid lesions, adenoma, and hyperplasia, appeared 100% hypoechoic in our study. As described in other studies, the fat tissue component reduction in these lesions is responsible for the aforementioned appearance, in the US [3,8]. Part of lesions presented with cystic inclusions, more commonly in PH (14.3% in PA vs. 58.2% in PH, *p* = 0.001). There was a similar tendency for hyperechoic central parts in both lesion types, although with no statistically significant difference in this study. Cakir et al. have also reported the majority of parathyroid lesions being hypoechoic (92.1% of typical PA, 100% of PH); with cystic inclusions more frequent in PH 17.6% vs. PA 13.9%, respectively [23]. Parathyroid gland cysts are reported to develop possibly due to local degeneration of the gland, following retention of colloid secretion or after haemorrhage [24]. Cysts may be seen in both malignant and benign parathyroid lesions [25,26,27]. Hyperplasias were generally smaller volume than adenomas in our study (*p* = 0.039); however, PH may present with the same size as PA, due to long-term dialysis in patients with secondary hyperparathyroidism (HPT) [28].

In our study, CD showed that vascularization type in parathyroid lesions may vary from central to combined, with the last being the most common vascularisation pattern. Feeding polar vessels was seen in the lesion periphery in up to 97.6% cases, a finding reported in other studies [29,30,31]. The presence of a polar feeding artery, a branch of the inferior thyroid artery, increases sensitivity by 10% and accuracy by 54% [31]. Thus, Doppler US imaging may help to distinguish suspected parathyroid lesions from other neck structures, such as the thyroid nodules (most commonly, peripheral vascularisation pattern) and lymph nodes (predominantly, hilar vascularisation) [32,33,34]. Additional US imaging modalities may help to distinguish between parathyroid lesions.

In the present study, PA had early hyper-enhancement that was more prominent in the rim of a lesion, with central wash-out in the later phases in CEUS. PH, presented with homogeneous enhancement, with similar median early contrast uptake time and early wash-out time and, in some cases, could not be distinguished from PA before quantitative analysis. Batista da Silva et al. have also reported persisting hypervascularisation in the PA periphery and central wash-out [35]. Ramirez et al. reported centripetal enhancement in parathyroid adenomas [36]. Authors of numerous studies reported different CEUS vascularization patterns between neck structures, which may be useful in differentiating parathyroid lesions from neck lymph nodes or thyroid lobules. Benign lymph nodes display centrifugal and homogenous enhancement [37,38,39]. Thyroid adenoma presents with homogenous hyperenhancement, unlike the thyroid gland in a “fast-in and slow-out” fashion, and may not have wash-out. Papillary thyroid carcinoma is characterized by inhomogeneous hypo-enhancement [40].

PA above 1000 mm^3^ (Group 2) had slower early uptake of contrast agent and slower wash-out than PA below 1000 mm^3^ (Group 1) in qualitative analysis. No significant difference was observed between Group 1 and Group 2 lesions when comparing ACSI, wash-in rate, and wash-out rate in patients who received 1 mL or 2 mL of intravenous contrast agent.

Other significant differences in PA and PH on quantitative evaluation—adenomas had increased contrast uptake in the periphery and early wash-out in the central part. However, hyperplasias had higher ACSI, WiR, and WoR in central and peripheral parts than adenomas (*p* < 0.001). The mean values of pooled ACSI results from PA showed a lesser difference in contrast uptake between centre and periphery that could be explained by the variability of the PA size, with larger volume PA presenting more evident differences.

There has been a controversy over the usefulness of CEUS in parathyroid imaging. Uller et al. reported that the sensitivity and specificity of CEUS were up to 98.4% in the diagnosis of PA [6]. Agha et al. study results showed that the sensitivity of CEUS for detection of PA, in the case of goitres, was significantly higher than that of the conventional US (100% vs. 64%, *p* = 0.005) and in cases after previous neck surgery (100% vs. 58%, *p* = 0.003) [41]. In the same study, using CEUS, double adenomas could be detected in all 5 cases, compared to conventional US (100% vs. 25%, respectively, *p* = 0.001). On the contrary, Karakas et al. concluded that no added value could be obtained using CEUS compared to conventional US and radionuclide imaging results in a limited cohort study [42]. Our study’s CEUS approach achieved a sensitivity of 90.9% and specificity of 72.7%, regardless of the contrast media volume use (1 mL vs. 2 mL). In comparison to the other studies, Li et al. used 1–2 mL of contrast agent and reported sensitivity up to 94.37%; other authors used 2.4 mL of contrast agent—Hornung et al., Ramirez et al., and Agha et al., achieved a sensitivity of 98.3%, 66.7%, and 97.1%, respectively [36,41,43,44]. We used average contrast signal intensity, wash-in rate, and wash-out rate parameters in post-processing analysis, as no definitive parameter or parameter group is defined in the quantitative evaluation of parathyroid lesions. After post-processing of acquired raw CEUS data using quantification software, the final specificity significantly increased up to 80%; however, sensitivity increased up to 98%. Batista da Silva et al. reported sensitivity up to 89.3%. They described persisting hypervascularisation in the periphery of PA in quantitative analysis (mean time to peak 7.93 s centrally, 8.36 s in the periphery, mean transit time 56.6 s centrally, 64.5 s in the periphery), followed by central wash-out [35]. Our results, with a qualitative assessment of relatively low specificity, could be explained by the study methodology. We included PH in our cohort, while others excluded hyperplasias from their analysis [35,36,41].

Sensitivity had decreasing tendency in smaller lesions (median 374 mm^3^) vs. larger lesions (median 916 mm^3^), as former lesions were harder to detect and analyse.

The sensitivity of 99mTc-MIBI scintigraphy alone was 68.4% (*p* = 0.7), but when combined with CEUS ultrasound, the sensitivity reached up to 94.3% (*p* = 0.03), as scintigraphy may detect lesions that could not be seen on ultrasound, e.g., in cases of atypical localisation—retrosternal or paravertebral.

Current cohort scintigraphy sensitivity results (68.4%) may be considered suboptimal, however, existing heterogeneity of the performed protocols, radiopharmaceutical activity in different institutions may affect this rate. Above mentioned and other factors have been thoroughly described in the meta-analysis that also shows a range of sensitivity of SPECT/CT scintigraphy 64–100% (95% CI: 84–92%) [45], and, therefore, we believe that our result of combined planar and SPECT/CT performance is appropriate. The evaluation of false-negative results may be interpreted ambiguously, especially in multiglandular disease, where there is a risk to locate one lesion and miss the other ones.

Despite the positive role of CEUS in the differentiation of adenoma from hyperplasia, we acknowledge the limitations of our study. We could not perform post-processing of acquired CEUS data in 31 cases out of 82 lesions, which can be explained by small lesion size, deep lesion localisation, and movement artefacts. Similar limitations have also been reported by another author group—the possibility to perform quantitative analysis in 28/42 cases [35]. One of the potential disadvantages of B-mode US and CEUS is the subjective interpretation by the operator since diagnostic criteria, such as early enhancement and onset of a wash-out, usually rely on the examiner’s interpretation. Therefore, it is essential to have a reliable quantitative evaluation for better reproducibility of the examination results. The present study was conducted in a single centre with a relatively low number of patients. To our knowledge, only a few studies are investigating the role of using quantification post-processing software for the interpretation of CEUS in parathyroid lesions [35,36]. Thus, additional prospective multicentre imaging studies are advised. Overall, quantitative software tools, for measuring time intensity curves during CEUS, could improve the diagnostic results and should, therefore, be considered for future routine use in patients that have a differential diagnosis between adenoma and hyperplasia, as well as in cases where adjacent lymph nodes may mimic parathyroid lesion.

## 5. Conclusions

CEUS of parathyroid lesions shows potential in the differentiation of adenoma from hyperplasia, regardless of the amount of contrast media injected. The quantitative analysis improved the specificity of differentiation between lesions by CEUS, and qualitative analysis improved sensitivity results. Hyperplasia was characterized by tendency of homogeneous enhancement, fast uptake, homogeneous wash-out appearance, and adenoma by peripheral uptake, central wash-out, and reduced hemodynamics. The use of CEUS quantification methods are advised to improve the ultrasound diagnostic role in suspected parathyroid lesions.

## Figures and Tables

**Figure 1 medicina-58-00002-f001:**
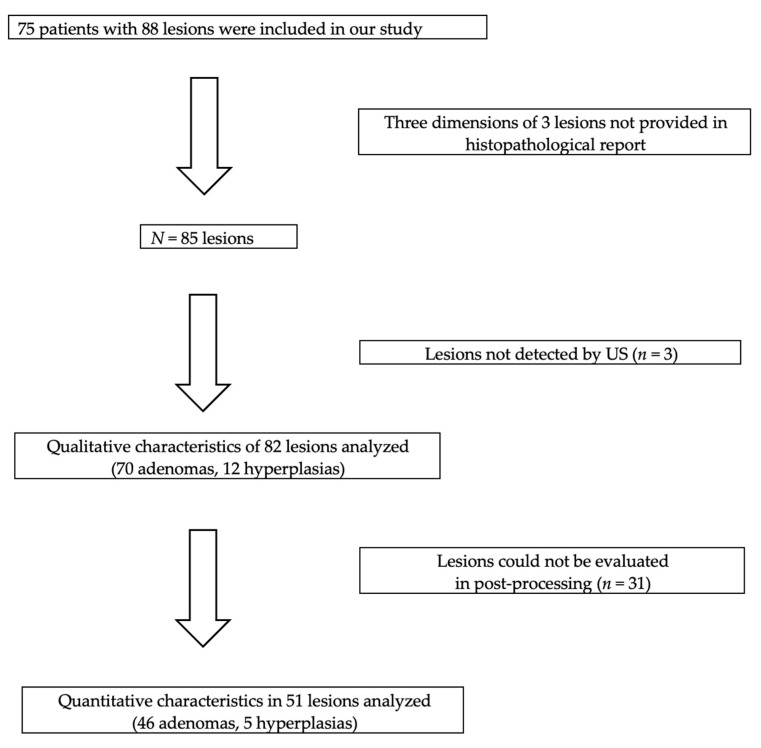
Flowchart of included patients.

**Figure 2 medicina-58-00002-f002:**
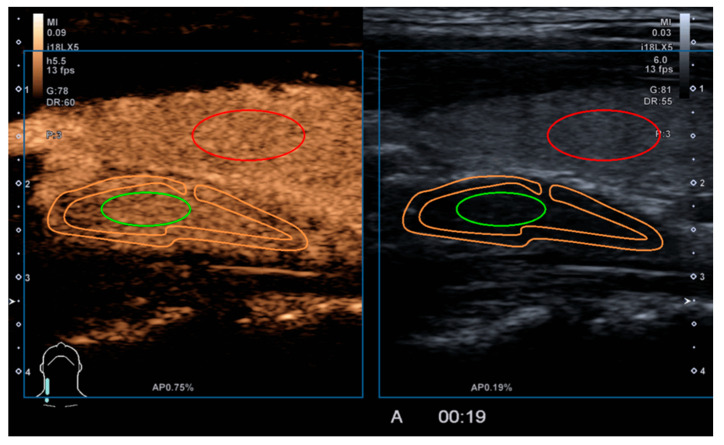
Post-processing of acquired CEUS examination. Marking the regions of interest in the parathyroid centre (green), parathyroid periphery (orange), thyroid gland (red).

**Figure 3 medicina-58-00002-f003:**
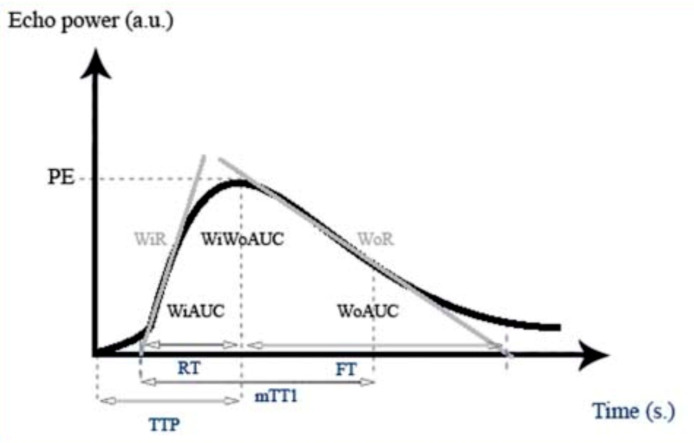
Signal intensity (echo power) in arbitrary units (a.u.), as a function of time in seconds (s), in a parathyroid lesion. Contrast enhancement increases until reaching Peak-Enhancement (PE). Wash-in area under the curve (WIAUC), Wash-in Perfusion Index (WiPI), Wash-in Rate (WiR), Wash-out area under the curve (WoAUC), Wash-in and Wash-out area under the curve (WiWoAUC), Wash-out Rate (WoR), Time to Peak (TTP), rise time (RT), mean transit time (local) (mTTl) and fall time (FT), permission to use the image is provided [20].

**Figure 4 medicina-58-00002-f004:**
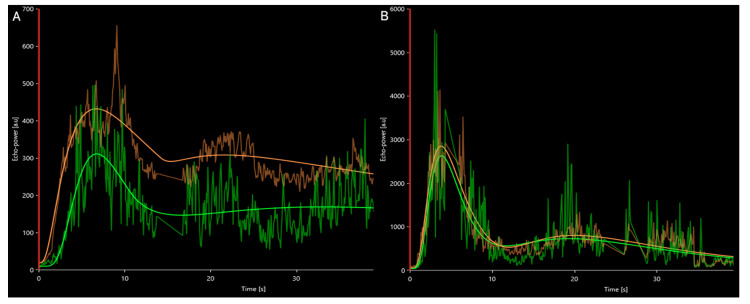
Time intensity curve of average contrast signal intensity in the parathyroid lesions, green—central part of the lesion, orange—periphery. (**A**) PA—higher ACSI values observed in periphery; (**B**) PH—contrast dynamics are similar in lesion centre and the periphery, with higher ACSI values than in PA. The ascending and descending parts of the curve were linear. A second small peak of enhancement, due to reperfusion was observed.

**Figure 5 medicina-58-00002-f005:**
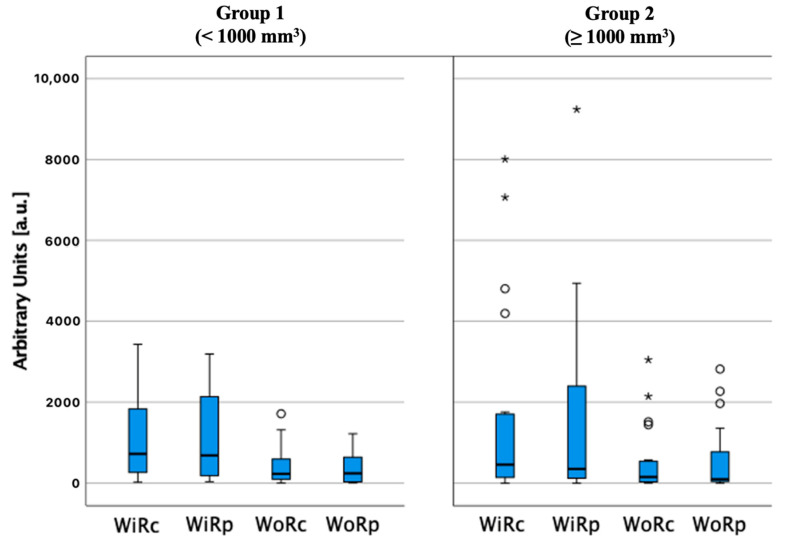
Wash-in and wash-out rate comparison in the parathyroid lesion centre (WiRc and WoRc, respectively) and periphery (WiRp and WoRp, respectively) in different parathyroid adenoma volume groups, ○ and *—number of cases (not shown).

**Figure 6 medicina-58-00002-f006:**
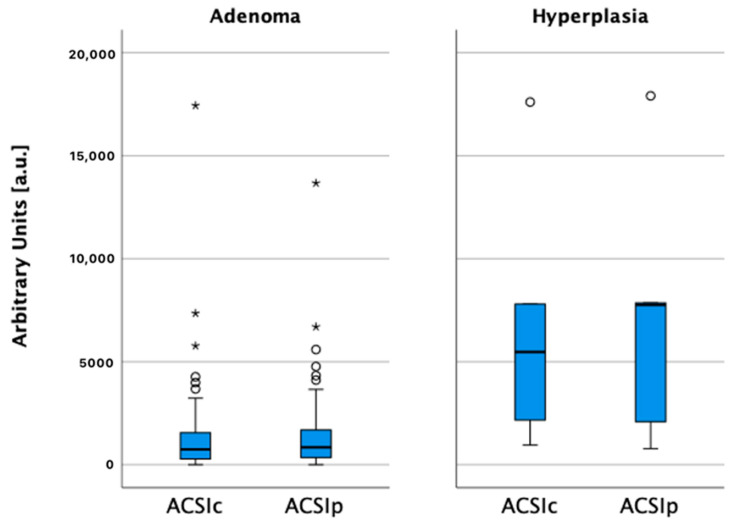
Average contrast signal intensity in central (ACSIc) and peripheral (ACSIp) parts of hyperplasias and adenomas, ○ and *—number of cases (not shown).

**Figure 7 medicina-58-00002-f007:**
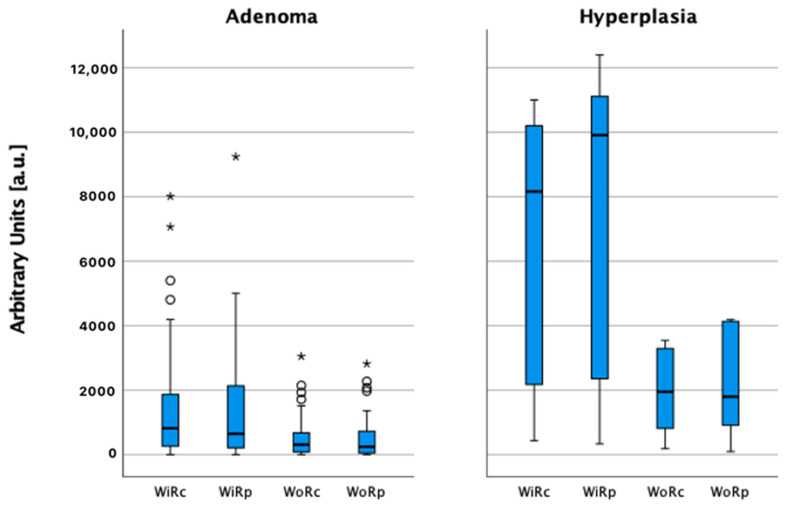
Wash-in and wash-out rate comparison in the parathyroid lesion centre (WiRc, WoRc, respectively) and periphery (WiRp, WoRp, respectively) in parathyroid hyperplasia and adenoma, ○ and *—number of cases (not shown).

**Table 1 medicina-58-00002-t001:** Baseline characteristics of patients.

Characteristics	Number of Patients (*n* = 75)
Age (years), mean (range)	59.8 (19–82)
Female	64 (85.3%)
Male	11 (14.7%)
Clinical data	
Fatigue	39 (52.0%)
Osteoporosis	25 (33.3%)
Kidney stones	23 (30.6%)
Morphology	Number of lesions (*n* = 88)
Adenoma	71 (80.7%)
Hyperplasia	17 (19.3%)
Characteristics	Number of patients (lesions)
Single lesion:	67 (67)
Adenoma	60 (60)
Hyperplasia	7 (7)
Multiple lesions	8 (21)
Double adenoma	4 (8)
Adenoma and hyperplasia	2 (4)
Four hyperplasias	1 (4)
One adenoma and four hyperplasias	1 (5)
	Total: 75 (88)

**Table 2 medicina-58-00002-t002:** Qualitative characteristics of 82 parathyroid lesions.

Characteristics	Adenoma (*n* = 70)	Hyperplasia (*n* = 12)	*p*-Value
Median (range), mm	13 (3–43)	10 (2–30)	0.995
Median volume (range), mm^3^	955 (65–8190)	364 (52–4680)	0.039
<1000 mm^3^—*n* (%)	36 (51.4%)	10 (80%)	
≥1000 mm^3^—*n* (%)	34 (48.6%)	2 (20%)	
Hyperechoic central part	31 (44.3%)	6 (50%)	0.717
Cystic component	10 (14.3%)	7 (58.2%)	0.001
Polar vessel	70 (100%)	10 (83.3%)	0.565
Vascularisation pattern			
Central	8 (11.4%)	1 (8.3%)	
Peripheral	19 (27.1%)	2 (16.7%)	0.924
Combined	33 (47.2%)	8 (66.7%)	
Absent	10 (14.3%)	1 (8.3%)	
Median (range)			
The onset of contrast uptake (s)	11 (4–20)	12 (7–20)	0.714
Time of maximum contrast concentration (s)	16 (8–30)	18 (11–28)	0.067
The onset of contrast wash-out (s)	27 (12–120)	25 (18–120)	0.553
Serum parathormone, pg/mL	162 (74–1166)	186 (90–1360)	0.026
Serum calcium, mmol/L	2.8 (2.3–4.2)	2.7 (2.6–3.1)	0.922

**Table 3 medicina-58-00002-t003:** Adenoma characteristics depending on lesion size (patients with single adenoma).

	Group 1	Group 2	*p*-Value
	<1000 m^3^ (*n* = 28)	≥1000 mm^3^ (*n* = 32)	
	Median (Range)	Median (Range)	
Onset of contrast uptake (s)	11 (8–20)	12 (7–18)	0.901
Time of maximum contrast concentration (s)	16 (12–30)	18.8 (9–30)	0.341
The onset of contrast	27 (15–120)	29 (15–120)	0.376
wash-out (s)			
Serum parathormone, pg/mL	131 (74–363)	182 (84–1059)	0.005
Serum calcium, mmol/L	2.7 (2.5–3.3)	2.8 (2.4–4.2)	0.199

**Table 4 medicina-58-00002-t004:** Post-processing parameters of PA comparison, with contrast media volume 1 mL vs. 2 mL.

	**1 mL**		**2 mL**		** *p* ** **Value**
	**Mean**	**95%-CI**	**Mean**	**95%-CI**	
ACSIc (a.u.)	1756	396–3174	1720	436–3005	0.386
ACSIp (a.u.)	1785	539–2613	1778	679–2877	0.436
WiRc (a.u.)	1146	530–2363	1474	684–2264	0.402
WiRp (a.u.)	1423	538–2308	1369	599–2140	0.436
WoRc (a.u.)	535	187–884	530	258–801	0.631
WoRp (a.u.)	532	173–892	471	208–734	0.356

Quantification toolbox (VueBox, Bracco, Milan), abbreviations: ACSI (Average Contrast Signal Intensity, c—centre, p—periphery); WiR (Wash-in Rate, c—centre, p—periphery); WoR (Wash-out Rate, c—centre, p—periphery); a.u.—Arbitrary Unit.

**Table 5 medicina-58-00002-t005:** Post-processing parameters comparison of parathyroid adenoma vs. hyperplasia.

	Adenoma		Hyperplasia		*p*-Value
	Mean	95%-CI	Mean	95%-CI	
ACSIc (a.u.)	1638	779–2366	6797	1416–15012	0.002
ACSIp (a.u.)	1646	909–2512	7275	1111–15662	<0.001
WiRc (a.u.)	1502	924–2081	8160	431–12356	<0.001
WiRp (a.u.)	1445	871–2019	7222	412–14031	<0.001
WoRc (a.u.)	558	345–770	1956	123–1956	<0.001
WoRp (a.u.)	552	310–735	2224	92–4542	<0.001

Quantification toolbox (VueBox, Bracco, Milan), abbreviations: ACSI (Average Contrast Signal Intensity, c—centre, p—periphery); WiR (Wash-in Rate, c—centre, p—periphery); WoR (Wash-out Rate, c—centre, p—periphery); a.u.—Arbitrary Unit.

## Data Availability

The data presented in this study are available on request from the corresponding author. The data are not publicly available yet due to ongoing research.

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
