# Peer review of "Contrast-Enhanced Ultrasound Qualitative and Quantitative Characteristics of Parathyroid Gland Lesions"

_medicina, 2021, doi:10.3390/medicina58010002_

Round 1

Reviewer 1 Report

Thanks for submitting good research. Your research will greatly influence the review and treatment of parathyroid nodules.

(1) Line 26:
There is no object in ‘we used 1ml or 2ml during the CEUS exam’. We do not understand what was used.

(2) Line 91: The sentence ‘samples were obtained since 2019’ is ambiguous. I would like to write a clear period. (ex. from May 2019 to June 2020)

(3) Figure 5:
I hope that the definitions for groups 1 and 2 that you wrote on lines 150-153 are also explained in the figure. The description of the picture is insufficient.

(4) Figure 6,7 :
  In Table 2, adenoma is on the left and hyperplasia is on the right. It's changed in the figure, so it's confusing. It would be nice to be unified with the table.

Reviewer 2 Report

In the manuscript, the authors deal with a problem of parathyroid adenomas and hyperplasia visualization which is a common problem due to high frequency of hyperparathyroidism.  The authors use the contrast enhancement USG for their visualization which has already been described in the literature several times, but is still an issue that can potentially improve the preoperative effectiveness of parathyroid imaging.

However there is several points which could be clarified:

  •       the authors choose the cohort among patients with parathyroid enlargement seen on USG, while the clinical problem results from the frequent lack of visualization of the parathyroid glands on ultrasound
  •      what was the overall sensitivity of CEUS in whole group of patients with hypercalcemia due to hyperparathyroidism
  •     it will be more valuable if the sensitivity of CEUS will be compared to the most frequently used method of parathyroid adenomas visualization, i.e. 99mTc-MIBI scintigraphy
  •     whether there was a relationship between PTH values and the image in CEUS
  •   which patients should undergo CEUS? All with parathyroidism? This with negative 99mTc-MIBI scintigraphy?
  •    was there any correlation between parathyroid diameter/volumes and sensitivity of CEUS?
  •    the normal ranges for s-Ca and s-PTH are not provided
  •    what you mean by term solitary hyperplasia?

Reviewer 3 Report

The study by Sergejs Pavlovics compares Color Doppler (CD), and contrast-enhanced ultrasound (CEUS) to traditional US is a struggle to answer the question whether these methods can be utilized by clinicians. Presented results confirm superiority od the aforementioned methods over the standard US.

There are still some questions remaining:

  • were the results compared to other imaging options (for example Tc-99m sestamibi parathyroid scintigraphy) by the authors or other researchers from literarture? If so, perhaps it would be worth to mention it in the study?

Minor comments:

1000 mm3 – please correct the mm3 symbol

Figure 5 – Group 1 and group 2 – please add description which group is which ( <1000mm3 and >1000mm3)

Round 2

Reviewer 2 Report

Thank you for answering the questions, but still there are some ambiguities in the text.

  • There is no information in the methodology whether scintygraphy with 99mTc-MIBI was done in all patients in your study group (as suggested in your replay) and which technique had you used
  • According to the literature, in meta-analyses the sensitivities of SPET/CT 99mTcMIBI scintigraphy is 84% - 88%  (Wei WJ, Shen CT, Song HJ, Qiu ZL, Luo QY. Comparison of SPET/CT, SPET and planar imaging using 99mTc-MIBI as independent techniques to support minimally invasive parathyroidectomy in primary hyperparathyroidism: A meta-analysis. Hell J Nucl Med. 2015 May-Aug;18(2):127-35. doi: 10.1967/s002449910207. Epub 2015 Jul 20. PMID: 26187212, Treglia G, Sadeghi R, Schalin-Jäntti C, Caldarella C, Ceriani L, Giovanella L, Eisele DW. Detection rate of (99m) Tc-MIBI single photon emission computed tomography (SPECT)/CT in preoperative planning for patients with primary hyperparathyroidism: A meta-analysis. Head Neck. 2016 Apr;38 Suppl 1:E2159-72. doi: 10.1002/hed.24027. Epub 2015 Jul 6. PMID: 25757222).  So, in a situation when to your analysis you include only  patients with parathyroid nodules seen in USG, it is expected to have even higher sensitivity of the scintigraphy than indicated in the meta-analysis. According to your data it equals only 68.4% - could you comment?
  • In addition, the SPECT/CT acquisition provides very accurate information about the location of the visualized parathyroid glands - why you suggest in discussion otherwise?
  • Finally, I do not agree with the opinion that CEUS should be performed in every patient with hyperparathyroidism, please include in discussion rational indications for such a test.
